# Indigenous Knowledge and Seasonal Calendar Inform Adaptive Savanna Burning in Northern Australia

**Michelle McKemey** [1,*] , **Emilie Ens** [2], **Yugul Mangi Rangers** [3], **Oliver Costello** [4] and **Nick Reid** [1]

1   Ecosystem Management, School of Environmental and Rural Science, University of New England, Armidale, NSW 2351, Australia; nrei3@une.edu.au
2   Department of Environmental Sciences, Macquarie University, 12 Wally's Walk, Sydney, NSW 2109, Australia; emilie.ens@mq.edu.au
3   Caring for Country Branch, Northern Land Council, GPO Box 1222, Darwin, NT 0801, Australia; Yugulmangi@nlc.org.au
4   Firesticks Alliance Indigenous Corporation, Rosebank, NSW 2480, Australia; ocostello@firesticks.org.au
*   Correspondence: michellemckemey@gmail.com; Tel.: +61-(0)437-350-597

**Abstract:** Indigenous fire management is experiencing a resurgence worldwide. Northern Australia is the world leader in Indigenous savanna burning, delivering social, cultural, environmental and economic benefits. In 2016, a greenhouse gas abatement fire program commenced in the savannas of south-eastern Arnhem Land in the Northern Territory, managed by the Indigenous Yugul Mangi rangers. We undertook participatory action research and semi-structured interviews with rangers and Elders during 2016 and 2019 to investigate Indigenous knowledge and obtain local feedback about fire management. Results indicated that Indigenous rangers effectively use cross-cultural science (including local and Traditional Ecological Knowledge alongside western science) to manage fire. Fire management is a key driver in the production of bush tucker (wild food) resources and impacts other cultural and ecological values. A need for increased education and awareness about Indigenous burning was consistently emphasized. To address this, the project participants developed the *Yugul Mangi Faiya En Sisen Kelenda* (Yugul Mangi Fire and Seasons Calendar) that drew on Indigenous knowledge of seasonal biocultural indicators to guide the rangers' fire management planning. The calendar has potential for application in fire management planning, intergenerational transfer of Indigenous knowledge and locally driven adaptive fire management.

**Keywords:** ecological calendar; Traditional Ecological Knowledge; cross-cultural; fire management; Indigenous fire; fire ecology; wildfire; wildland fire; Indigenous

## 1. Introduction

Wildfire management is an escalating issue globally, with economic, environmental, social and cultural consequences [1–4]. In fire-prone regions such as the Americas, Australia and parts of Asia and Africa, wildfire management presents a formidable ongoing challenge that must be urgently addressed. Wildfire is also a growing problem in regions where it has not previously been a priority, such as southern Europe [5].

Indigenous peoples have been effectively managing fire on their ancestral estates for millennia. Beginning tens of thousands of years ago, hunter–gatherers around the world used fire to reduce fuels and manage wildlife and plants [6]. Australian Aborigines, who have inhabited Australia for sixty-five thousand years [7], maintained a complex system of land management using fire and the life cycles of

native plants to ensure plentiful wildlife and plant foods throughout the year [8]. Fire management is driven by an Indigenous group's cosmovision, encapsulated in Australia's First Nations peoples' term "caring for country," whereby the maintenance and restoration of land and ecosystems is inextricably linked to human wellbeing, spirituality, kinship systems and culture.

In many nation-states following the onset of colonization, traditional Indigenous fire management practices were disrupted through policies such as the removal of Indigenous peoples from their lands, prohibition of traditional practices and fire exclusion [9,10]. In recent decades, recognition of Indigenous knowledge and practice of fire management has led to the re-emergence of several Indigenous fire management programs [11–17]. For example, in South America, following a period of wildfire suppression and prohibition of Indigenous fire practices, Indigenous groups and government agencies are working towards a participatory and intercultural fire management approach [18]. While there is growing recognition of the important role Indigenous peoples could play in managing natural hazards on their ancestral lands, internationally, collaborative decision making is often considered to be "an aspiration more than a reality" [19,20].

Northern Australia was colonized by Europeans from 1860 onwards, leading to the depopulation of ancestral Indigenous clan estates ("country") and subsequent disruption to traditional fire regimes in many areas [21]. Prior to colonization, fire was managed by Indigenous custodians through culturally driven, systematic, patchy, landscape-scale burning of country. Following colonization and disruption of traditional Indigenous fire practices, fire regimes became dominated by extensive wildfires occurring predominantly during the severe fire weather conditions of the late dry season, covering many thousands of square kilometers and causing significant environmental damage [22–24].

From the 1990s, the establishment of community ranger groups and Indigenous Protected Areas (IPAs) facilitated many of Australia's Indigenous peoples to re-connect to, and participate in, land management activities [25]. IPAs are areas of land and sea managed by Indigenous groups as protected areas for biodiversity conservation through voluntary agreements with the Australian Government. In 2020, IPAs contributed over one hundred million hectares, or 54%, of the National Reserve System of Australia [26].

From 1997, Indigenous people, fire researchers, fire management authorities and public and private funding agencies began to promote active restoration of customary Indigenous fire management in northern Australia [22]. From 2006, Indigenous land management organizations commenced formal agreements with industry and government partners to off-set greenhouse gas (GHG) emissions through "savanna burning."

Savanna burning is described by the Australian Government [27] as the "savanna fire management—emissions avoidance method [that] credits activities that reduce the emission of greenhouse gases from fire in savannas in northern Australia, through a reduction in the frequency and extent of late dry season fires." Tropical savanna ecosystems account for around 22% of the global land surface [28] and 50% of the total annual biomass is burned globally [29]. The seasonally dry tropics of northern Australia account for 12% of the world's tropical savanna biome and have global significance. In these ecosystems, fire is "arguably the greatest natural and anthropogenic environmental disturbance" [30] and the most important tool available to Indigenous peoples to manage country [24,31].

By 2019, there were over 70 savanna burning projects across northern Australia [24], while Indigenous savanna burning projects in Arnhem Land (Figure 1) alone earned approximately $10M AUD per annum [32]. These projects provide Indigenous rangers in remote, socio-economically disadvantaged communities with an opportunity to care for their country and earn income to support community development. For example, the Warddeken IPA Indigenous rangers' savanna burning program, using traditional and contemporary practices, has generated substantial revenue as a result of carbon offset sales. This program has also lead to socio-economic and cultural outcomes, such as increased employment, skills, confidence, health, wellbeing, pride, maintenance of culture and Indigenous languages, and greater respect for Traditional Ecological Knowledge (TEK) [33]. Ansell and

Evans [24] found that Traditional Owners participating in the broader Arnhem Land Fire Abatement (ALFA) project want to continue healthy fire management on their country, see fewer wildfires, protect biodiversity and culturally important sites, maintain and transfer knowledge, as well as create a carbon abatement. They found that, whilst annually variable, the savanna burning projects are meeting these goals. Due to these successes, Australia's Indigenous savanna burning program is considered a world leader, with interest from peoples in savanna regions globally to instigate similar programs [34].

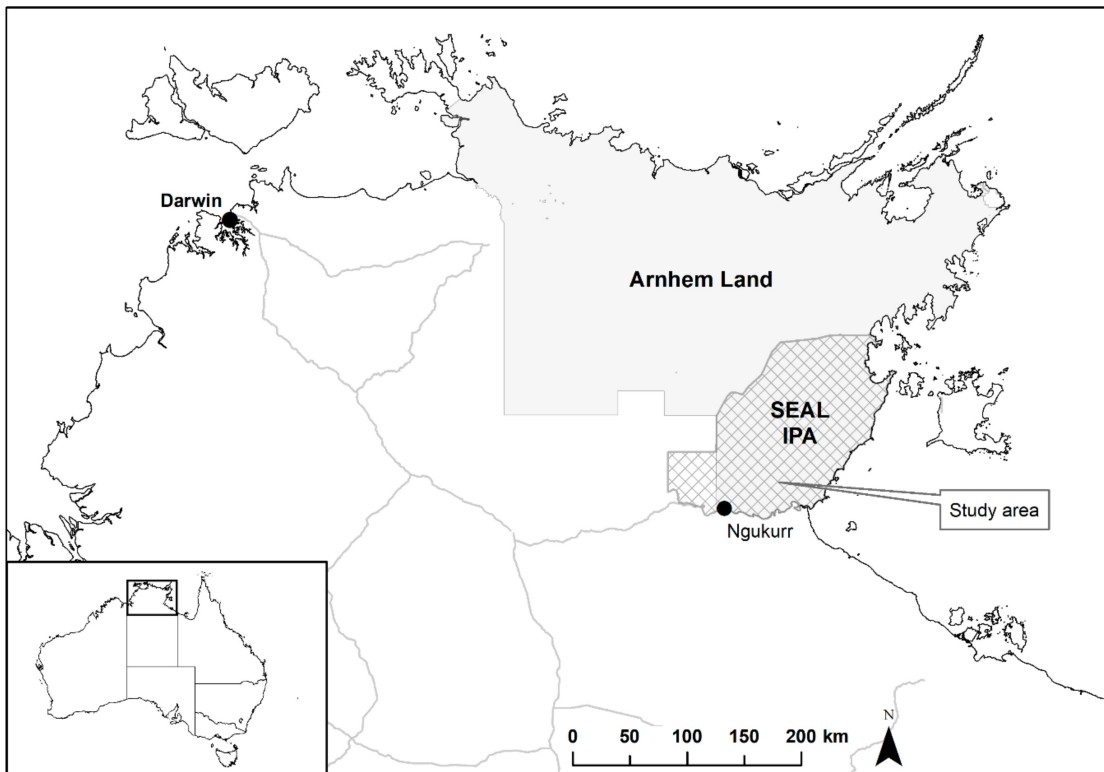

**Figure 1.** South East Arnhem Land Indigenous Protected Area (SEAL IPA), Northern Territory, Australia.

Indigenous savanna burning projects aim to use cross-cultural science (Indigenous and western knowledge systems [35]) for best-practice adaptive management of fire, incorporating natural and cultural values. Indigenous knowledge and languages are under grave threat globally [36] and intergenerational transmission of this knowledge is constrained by pervasive variables in the contemporary environment [37,38]. Around the world, seasonal calendars are viewed by Indigenous communities as an important way to collect and share Indigenous knowledge [39] and to signify the profound connection between people, country and the yearly cycle of change on country [40]. These calendars provide a graphical representation of Indigenous seasonal knowledge, including knowledge of the weather, ecological indicators, seasonal cycles of plants and animals, and their links with Indigenous culture and land uses [41].

Indigenous seasonal calendars have been used in the monitoring and adaptive management of natural resources, agricultural systems [42–46], climate change [47–49], water [50] and fire regimes [11], and to guide eco-health decision making [51]. In Australia, many Indigenous seasonal calendars have been developed, based on phenological observations of local environments, and are often linked to practices such as harvesting of traditional resources and fire management [52,53]. For example, some seasonal calendars depict ecological indicators or specific seasonal conditions as a guide for fire management [53,54], although this information is often not detailed for the primary purpose of the seasonal calendar. While seasonal calendars are generally popular for cross-cultural interpretation of Indigenous ecological knowledge, Prober et al. [41] and Franco [55] noted that seasonal knowledge is underutilized in natural resource management.

Our review of the scientific literature reveals that fire management informed by Indigenous knowledge is sophisticated, effective and widely underutilized across the globe. However, processes for sharing and understanding Indigenous knowledge are limited, and could contribute to improved management of social–ecological systems. In this study, we attempted to address this knowledge gap by sharing Indigenous knowledge of fire, and developing a fire and seasons calendar to improve adaptive fire management and communication. The objectives of our study were: to describe Indigenous fire knowledge and practice in South East Arnhem Land (SEAL) IPA (Figure 1); to explore how fire management affects cultural and ecological values; to use local Indigenous knowledge to develop a fire and seasons calendar for the SEAL IPA to guide savanna burning; and, to outline how the fire and seasons calendar can be applied to improve adaptive fire management and communication in northern Australia and other fire-prone regions and countries around the world.

## 2. Materials and Methods

### 2.1. Study Area

Northern Australia is one of the world's rare, internationally significant, large natural areas, comparable to wilderness zones such as the Amazon rainforests, the boreal conifer forests of Alaska, and the polar wilderness of Antarctica [56]. Arnhem Land was declared an Aboriginal Reserve in 1931, and covers an area of 97 000 km$^2$. Due to its remoteness, Arnhem Land has retained many of its natural and cultural values, and is considered a refuge for Indigenous culture and native ecosystems, which have faced threats and significant degradation elsewhere.

The SEAL IPA was declared in 2016 covering an area of 19 170 km$^2$ on the western edge of the Gulf of Carpentaria in the Northern Territory and conserving extraordinary natural and cultural values [57]. The Indigenous Yugul Mangi (meaning "all of us," i.e., Indigenous people of South East Arnhem Land) Rangers were formally established in 2001 and manage the southern region of SEAL IPA in partnership with the Numbulwar Numbirindi Rangers.

The Yugul Mangi Rangers commenced the South East Arnhem Land Fire Abatement 2 (SEALFA2) savanna burning project in 2016, with the intention that "this initiative will provide enough income for our rangers to continue proper management of fire throughout our IPA" [57]. "The SEALFA2 project applies strategic early dry season burning activities to reduce the total area that is burnt each year and to shift the seasonality of burning from [the] late dry season to early dry season [of the monsoonal tropical climate]. This reduces emissions because the fires are less intense and burn less country each year" [58].

### 2.2. Qualitative Eco-Cultural Research

Qualitative ecological and cultural (eco-cultural) research was undertaken in the SEAL IPA during June 2016 and June 2019 (University of New England Human Ethics approval HE14-182 & 19-068, Northern Land Council Research Permit 86574). The Yugul Mangi rangers and SEAL IPA Elders were selected as collaborators due to their knowledge and practice of fire management. All participants resided in the town of Ngukurr (Northern Territory, Australia, Figure 1) at the time of the interviews and represented nine traditional Indigenous language groups from the SEAL IPA region. Ngukurr is a remote Aboriginal community with a population of 1149 people [59], although the population fluctuates as people move around among "outstations" (isolated houses located on traditional clan estates) and other towns.

Twenty-one Indigenous participants (eight female, 13 male) were interviewed, ten of whom (four female, six male) were interviewed in both 2016 and 2019 (Table 1). The number of participants was limited by the low total number of Elders and rangers and the availability of participants during field work. Participants were coded according to gender (male [M] or female [F]) and role (Elder [E] or ranger [R]). Younger community members also participated in order to learn from the Elders and develop skills in undertaking research.

**Table 1.** Participants in 2016 and 2019.

| Gender and Role [A] | Interviewed in 2016 | Interviewed in 2019 | Interviewed in both 2016 & 2019 | Total Participants (*n*) |
|---|---|---|---|---|
| FE | 4 | 5 | 3 | 6 |
| ME | 5 | 6 | 4 | 7 |
| FR | 1 | 2 | 1 | 2 |
| MR | 5 | 3 | 2 | 6 |
| TOTAL | 15 | 16 | 10 | 21 |

[A] F = female, M = male, E = Elder, R = ranger.

Following an inductive approach, research methods included participatory action research, semi-structured interviews and focus groups in a meeting forum [60], conducted during field work in Ngukurr and SEAL IPA. For the participatory action research, a non-Indigenous scientist (MM) and the Yugul Mangi rangers participated in ground-based savanna burning in the SEAL IPA in order to collect information and photographs for the fire and seasons calendar.

Semi-structured interviews were conducted singly or in groups of up to four Elders or rangers. Interview questions were based on the themes of traditional and contemporary Indigenous fire knowledge and management; cultural values of fire, plants and animals; resource use; weather conditions, seasonal changes, biocultural indicators, and use of the fire and seasons calendar (Appendix A). To facilitate discussion, information from five pre-existing seasonal calendars developed in 2000 by speakers of Roper River Kriol ("Kriol", the local Aboriginal language), Ritharrŋu/Wägilak, Ngandi, Marra and Wubuy (Nunggubuyu) languages were used [61–65]. Interviews were recorded audio-visually and transcribed. Interviews conducted in Aboriginal languages were translated into English with the assistance of local community members. Interview results were coded *a posteriori* manually and grouped into themes [66].

Focus group meetings were held at the Yugul Mangi rangers' headquarters and concentrated on working through blank calendar templates to fill in fire management activities and observations of seasonal changes throughout the year. A draft of the fire and seasons calendar was developed, and follow-up consultation was undertaken to provide feedback on the calendar. Senior language speakers at the Ngukurr Language Centre contributed to the calendar. The Indigenous rangers and non-Indigenous scientists worked together to finalize the calendar and develop the manuscript for publication.

## 3. Results

### 3.1. Traditional Indigenous Fire Management

Traditionally, burning was an important activity that was done purposefully by "the old people" (Indigenous ancestors). Reasons for burning included: to renew plants and grass, open up country, facilitate hunting, encourage bush tucker, communicate (signal between groups), improve safety, for cooking, to set up camp and for ceremony. Skills and knowledge about fire management were passed on from the old people orally and learned experientially. Some of the tools for traditional fire management included fire-drills, rocks, flint stone, firestick, dilly bag, paperbark (*Melaleuca* spp. bark) and pandanus (*Pandanus spiralis*) leaf. More recent tools include steel or knife, tin and matches. Burning was usually undertaken on foot, often when groups of people were out hunting. "*Mainly traditional burning was straight after the rain. When the rain stops the best time that we learnt to light fire is to go out with the old people hunting and they told us, burn here and burn there, it is only small patch burn*"—MR.

### 3.2. Contemporary Indigenous Fire Management

Contemporary Indigenous savanna burning uses cross-cultural science for best-practice management, combining traditional and Indigenous knowledge with western science. "*We are*

*showing, not even Australia, but worldwide that we are managing our Country with the knowledge that has been passed down from generation to generation by our grandfathers and ancestors, and still is. Because we got modern* [tools], *like using helicopter to burn, all season matches, drip torch. We are still using our same knowledge—when to burn, how to burn . . . We are doing the burning because our grandfathers did it before. Now they are gone, they taught us and we take note. Now we are teaching our children, our future rangers, we are teaching them to take over our ranger program, including fire. Fire management in our ranger program is very important because fire brings back life"*—MR.

Similar to traditional uses, fire is currently used for hunting, campfires, cooking, clearing around camp, to renew plants, ceremony, intergenerational knowledge transfer and to follow in the footsteps of the ancestors. More modern uses for fire also include: to earn income from carbon farming, to offset the burning of fossil fuels, to stop intense wildfires, to protect infrastructure, fire-sensitive areas and neighbors, and *"to show Australia and the world we know how to burn using traditional knowledge and modern technology"*—MR.

Contemporary burning uses both traditional and modern methods (Figure 2). Drawing on tradition, practitioners sometimes light fires the cultural way with sticks or paperbark, undertake burning walks where they follow traditional routes of the ancestors and light up the country, and follow cultural protocols (e.g. kinship system) for land management. Burning is undertaken by community members, Elders and rangers. The right people must light the fire on clan estates and rangers must have permission from the right people before they can burn the land. *"When we set up our ranger program it was set up with traditional kinship relations to the country. To have the right people to light up the fire"*—MR. Intense ("hot") fires and not following cultural protocols can cause conflict in the community. One of the rangers explained their relationship with the Traditional Owners: "[We (rangers) tell the Traditional Owners] *this is your land, you have every right to tell us what to do, where to burn and where not to burn. That is showing them that they have got authority for the land, not us, the ranger group, it is them. It makes us feel happy that we are taking them out, and to learn more of our culture and the way our ancestors and our forefathers have burned"'*—MR.

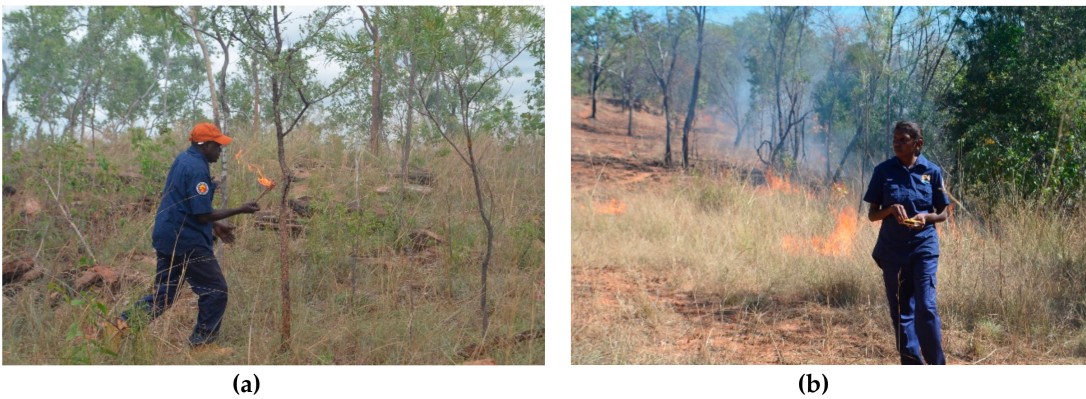

|            (a)            |            (b)            |

**Figure 2.** Yugul Mangi rangers lighting fires with (**a**) fire stick and (**b**) matches (photographs: Michelle McKemey).

Current methods employ ground and aerial burning. Tools for burning include: lighters and matches, drip torches, four-wheel-drive vehicles, helicopters and an aerial incendiary (Raindance) machine. Technology such as CyberTracker, iTracker, GPS, satellites and camera trapping is used for monitoring. Training, support and information are provided by other ranger groups and through organizations such as Arnhem Land Fire Abatement (ALFA), NT Bushfires Council, Northern Land Council, Charles Darwin and Macquarie Universities, and the North Australia Fire and Rangelands Information (NAFI) website. Remote-sensed data have proved useful to improve fire management practices [67] and remotely-sensed fire-scar maps are used to plan and review fire management. Contemporary burning generally begins in April and May (early dry season) and continues through to

the end of July. *"We are doing cool burning now* [June], *right time burning'*—FR. *'The more time you burn when it is cooler, the more carbon credits you get"*—MR. Some patch burning is undertaken in August and September to "clean up" areas that weren't burned during the cool period. Factors influencing burning include: type of landscape/ecosystem (e.g., hills or savanna), weather (especially amount and timing of rainfall), fire history (when an area was last burnt), fire-sensitive areas (such as bush tucker areas, rainforest, rock art, sacred sites and sandstone escarpment) and infrastructure (outstations, camping and recreational areas, property boundaries). Good fire is considered to be low-intensity ("cool") fire which burns slowly, only burns the grass not the trees, allows animals to escape and is generally conducted during the colder months of April to July. *"The right time to burn is when it is cool and the fire burns slowly . . . when we have heavy dew in the morning and the afternoon . . . The wind picks up usually midday and then we burn and that helps us push the fire along'*—MR. *'The time to stop burning is when it gets very hot"*—ME. "Nugudwan faiya" (Kriol for bad fire) are hot fires that burn the canopy, leave fire scars and black ash on the trunks of trees, burn in an arrowhead and are often ignited by lightning. *"A hot burn, that is a 'nugudwan faiya', that makes us unhappy. That is not right. Not only will the trees get burnt, it will stop the birds* [wanting] *to make their nest in there . . . and even kill small lizards—the skink that climbs up the tree—kills the landscape"*—MR.

## 3.3. Savanna Burning

The Yugul Mangi rangers formally commenced savanna burning in 2016. During the first three years of operation, participants shared that savanna burning has:

- provided income for the rangers to buy work-related resources, including gators, trailer, vehicles, fuel, helicopter time, firefighting uniforms, drip torches, matches, Raindance machine and pay for casual employees
- supported rangers in their fire management by improving engagement with western technology such as GPS, helicopters and satellite mapping whilst still using Indigenous ways of burning
- reduced wildfires by increasing early dry season burning.

A ranger explained how they combine traditional burning with modern technology: *"It works by us doing traditional burning with our new western technology . . . Through the Indigenous way we know which part of the area that we come to burn and the right time of burning"*—MR. Other rangers explained the economic, cultural and environmental benefits: *"Savanna burning benefits the country and the ecosystems. Country is healthy. Brings in dollars for us to use in our rangers for more resource"*—MR. *"The carbon money helps us to get more casuals, and fuel to take Elders on the chopper* [helicopter] *to look and to visit their Country"*—FR. However, many of the Elders interviewed had little understanding of savanna burning for GHG abatement, its application in the SEAL IPA or the income derived from this activity, as illustrated by the quote: *"What is carbon?"*—FE. There was also a mixture of positive and negative sentiments expressed by various participants regarding the outcomes of savanna burning. Negative comments focused on: too much or "wrong way" burning; detrimental impacts on bush tucker; lack of opportunity for community members to burn; mis-managed fire leading to wildfire; lack of education for young people, and that community members did not respect the work of the rangers. Positive comments focused on: continuing improvement in ranger management of fire; Indigenous Traditional Owners' support of rangers; increased income and resources generated from savanna burning; effective use of cross-cultural science; involvement of community members, Elders and young people in burning programs and their emotional wellbeing related to Indigenous fire practices; the potential for "right way" burning to restore important bush-tucker populations; rangers' respectful engagement with Traditional Owners; the community's respect for the work of the rangers, and community members having the opportunity to undertake burning practices.

### 3.4. Bush Tucker

Bush tucker, including edible plants and animals used for food that are sourced wild from the bush, was emphasized as important by most participants. The various resources utilized in different seasons were described and subsequently portrayed in the *Yugul Mangi Faiya En Sisen Kelenda*, the Roper River Kriol name for the Yugul Mangi Fire and Seasons Calendar (Figure 3).

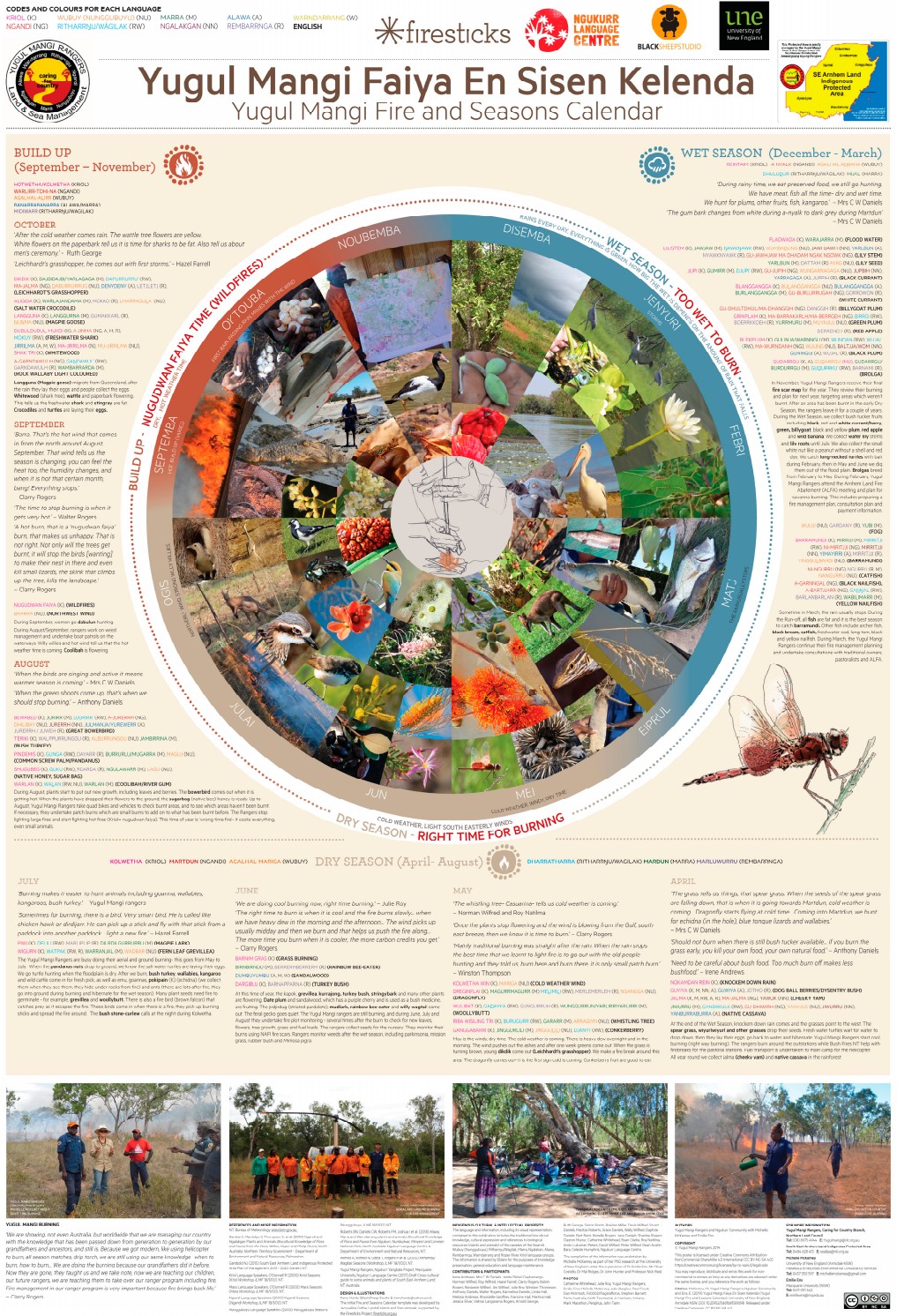

**Figure 3.** *Yugul Mangi Faiya En Sisen Kelenda*, the Yugul Mangi Fire and Seasons Calendar © Yugul Mangi Rangers [68] (CC BY-NC-SA 4.0).

Participants said that fire can impact on bush tucker by damaging and reducing the yield of native fruits or ground plants such as tubers. *"Any tree that bears berry, we usually keep the fires away from them. Or do a small fire burn so when the hot fires come, it doesn't cook them. It is stopped by the firebreak that we make"*—MR. Fire was said to be an important tool to protect and encourage growth of bush tucker resources. *"Burn-offs are important to keep our fruit, our shrubs, all the things that are in the bush that we use, for medicine or eating, so they can produce at the right time, especially with the berries and the yams"*—FE.

Many participants stressed the importance of careful burning around bush tucker plants, while some expressed concerns that burning too much, or at the wrong time, was having a detrimental impact on bush tucker abundance: *"Should not burn when there is still bush tucker available . . . if you burn the grass early, you kill your own food, your own natural food"*—ME. *"Need to be careful about bush food. Too much burn-off makes less bush food"*—FE. Fire can also affect traditional animal food resources by reducing feed for prey animals, providing 'green pick' for herbivores or improving hunting conditions. *"Burning makes it easier to hunt animals including goanna, wallabies, kangaroos, bush turkey"*—MR.

Participants identified a number of impacts (other than fire) on bush-tucker production, including rainfall, feral animals, weeds and non-customary harvest of resources. *"We thought burning was interrupting the bush tucker but it is mainly feral animals. A little bit burning but mainly ferals. We are really concerned about this"*—FE. *"Most* [bush-tucker production] *depends on the rain, too; the past years we haven't had enough rain, that's why. The rain, the watering of our plants comes from the rain"*—FE. Protection and regeneration of bush tucker was important for ecological and cultural values, and closely linked to fire regimes: *"The burning and the bush tucker and the country, it all comes together as normal practice . . . if you burn the right way, you look after the bush tucker, the bush plants"*—MR.

## 3.5. Biocultural Indicators

We define biocultural indicators as predictable, obvious, seasonal events that are culturally significant. Nineteen participants shared knowledge of 40 biocultural indicators related to weather conditions, resource use, fire management and cultural and spiritual events. *"They have different flowering season for all the trees. Different animals that we can hunt for. So one group of trees, one species, that flowers during that certain month then we know which animals are ready to be hunted. Other trees flower then you know different animals ready to be hunted. All the trees finish flowering then we know wild honey time."*—MR. Participants explained that some biocultural indicators are used as cues to initiate or cease certain fire management practices (Table 2, Figure 3). The rangers explained that reading country and noticing biocultural indicators is important when making decisions related to fire management: *"We look at the land first before we actually burn. What state is it in? Is it ready to be burning, or not? And you can tell that by the different color on the grasses. If it's brown enough, yeah well, it's good. And then if it's too green, we might have to miss a month"*—MR. *"They* [old people] *used to know when to burn and then when it comes like this now, this season. May, that's when they start off. We knew that cold weather coming up. Cold weather and we know when you already burnt and that dew and fog going to come . . . Green grass, new leaves and what we eat from the bush, like berries, plum and then the animals too, they follow where the best feed is. And then we know"*—MR.

**Table 2.** Yugul Mangi seasons, biocultural indicators and fire management practices.

| Season, including Indigenous Language Names [A] [69] | Weather Conditions [70] | Ranger and Elder Observations & Fire Management | Examples of Biocultural Indicators | Savanna Burning Practices |
|---|---|---|---|---|
| Wet season (approximately December to March) *Reintaim (Kriol, P1) A-nyalk (Ngandi, N90) Agalhal-anbana (Wubuy [Nunggubuyu], N128) Dhuludur (Ritharrŋu, N104, Wägilak, 106) Mijal (Marra, N112)* | Mean rainfall 132–183 mm/month. Mean maximum temperature 34–38 °C. Mean wind speed (9 am) 5.2–5.5 km/hr, (3 pm) 8.4–10.4 km/hr. Mean relative humidity (9 am) 65%–78%, (3 pm) 43%–57% | Too wet to burn. Rains every day, everything is green. How big the "wet" (period of rainy weather) is depends on the amount of rain that falls | *"Leichhardt's grasshopper [Petasida ephippigera], he comes out with first storms"*—FE. *"At the end of the Wet Season, knock 'em down rain comes and the grasses point to the west. The spear grasses, wiyurlwiyurl, and other grasses drop their seeds"*—MR | Yugul Mangi rangers receive their final fire scar map for the year, review their burning, plan for next year and attend the ALFA meeting. They consult Traditional Owners and neighbors regarding their fire plans |
| Dry season (approximately April–August) *Kol wetha (Kriol, P1) Martdun (Ngandi, N90) Agalhal-mariga (Wubuy [Nunggubuyu], N128) Dharratharra (Ritharrŋu, N104, Wägilak, 106) Mardun (Marra, N112) Marluwurru (Rembarrnga, N73)* | Mean rainfall 1–56 mm/month. Mean maximum temperature 30–34 °C. Mean wind speed (9 am) 5.2–7.1 km/hr, (3 pm) 11.8–14.1 km/hr. Mean relative humidity (9 am) 62%–74%, (3 pm) 29%–45% | Right time for burning. Cold weather, light south-easterly winds | *"When you look at that dragonfly, he start flying, you know cold weather time"*—FE. *"The whistling tree [Casuarina spp.] tells us cold weather is coming"*—ME. *"Once the wind is blowing from the Gulf [of Carpentaria], south-east breeze, then we know it is time to burn"*—MR | Yugul Mangi rangers start cool burning ("right-way" burning). The rangers burn around outstations and prepare firebreaks with neighboring pastoral stations. Fuel transport is undertaken to main camp for helicopter. From May to July, ground and aerial burning is undertaken. Yugul Mangi undertake monitoring at fire plots (several times after burn to check for new leaves, flowers, tree growth, grass and fuel loads) and use satellite technology to monitor fire scars |
| Build up (approximately September to November) *Hotwetha (Kriol, P1) Warlirr-tdhi-na (Ngandi, N90) Agalhal-alirr (Wubuy [Nunggubuyu], N128) Midiwarr (Ritharrŋu, N104, Wägilak, 106)* | Mean rainfall 2–42 mm/month. Mean maximum temperature 35–39 °C. Mean wind speed (9 am) 6.2–6.9 km/hr, (3 pm) 11.8–16 km/hr. Mean relative humidity (9 am) 53%–59%, (3 pm) 25%–30% | "Nugudwan faiya" (Kriol for hot, destructive fire) time- wildfires. Dry, hot weather time | *"Barra. That's the hot wind that comes in from the north around August, September. That wind tells us the season is changing, you can feel the heat too, the humidity changes, and when it is hot that certain month, bang! Everything stops"*—MR. *"We traditionally finish burning in September, when we see the dark clouds come in, that's the build-up, we stop then"*—MR. *"We know hot weather when we see that willy [whirlwind] … all the [magpie] geese [Anseranas semipalmata] they are fat now and it is right to go and hunt them"*—FE. | Yugul Mangi rangers stop lighting large fires and start fighting hot fires. This time of year is "wrong time fire"—it cooks everything, even small animals |

[A] Indigenous language names for seasons are displayed as Language name for season (Indigenous language name and AIATSIS code [69]).

## 3.6. Use of the Fire and Seasons Calendar

Many participants agreed that the calendar would be useful for young people, community members, other ranger groups, researchers and the public. *"It is good for our young ones to know, our way of looking after the country using fire. The calendar is the cycle of our ecosystem, the way we live and the way we hunt. Especially hunting and gathering food"*—MR. *"I think this calendar will be very, very useful for our children, to have them to learn about the different seasons and what is available during those seasons and most importantly, it is the burn-offs. What time the burn-offs should be done, that is the most important thing"*—FE.

Some participants felt that many people living in the SEAL IPA do not know how to burn properly, and need to be educated: *"Some adults have been brought up—going away from here to colleges, schooling outside of here—and hardly getting the knowledge that they should about the land and what their land would provide at what times, so that's important to all age groups"*—FE. They stated that the calendar could be

used at "culture camps" on country to teach young people, while the plant and animal photos in the calendar could help Elders to remember stories to share. This could be followed up with experiential learning of fire management on country: *"We would like to use the fire and seasons calendar as part of our program, to get the kids connected on country. To pass on knowledge about the environment and burning . . . We want to educate our kids for early burning so they don't do wrong way fire"*—FE. *"Using paper, that's a good help for the new generation, once they look at that and see what we've done, that's our seasonal calendar they will think that's what they do, on those months, until the end of the year. They don't learn that in school. So what we do is we show the kids that and when they come out on culture camp, some of that will be in their head, 'oh yeah I have seen that on the calendar chart,' they will say that, put it in black and white first to them and have that activity after, in real life"*—MR.

Rangers were enthusiastic to share the calendar with other Indigenous ranger groups: *"I would love to show* [the calendar] *to the ALFA group at my next meeting. Even our annual fire meeting where all the ranger groups come together in one location and do our planning. Each ranger group presents what they will do for the year, what they have done last year, we can present that too at our annual fire meeting"*—MR. One ranger said that the calendar would be particularly meaningful as it had been developed by the Indigenous rangers with assistance from non-Indigenous scientists, rather than having been developed in isolation from the community: *"This one* [calendar], *that's the ranger group doing our seasonal chart with you* [non-Indigenous scientist], *making a lot of difference"*—MR.

## 4. Discussion

Our study has described a contemporary system of Indigenous burning of country that uses Indigenous knowledge, traditional kinship systems and modern technology for best-practice, adaptive management of fire. The Indigenous Elders and rangers explained a fire management system that is based on intricate Traditional Ecological Knowledge (accumulated over thousands of generations of Indigenous people managing their clan estates) [71], local knowledge (of people living in and managing their environment on a day-to-day basis) and western science. This combined knowledge, coupled with Indigenous fire practitioners' ability to read country and identify biocultural indicators, places them in the unique position to be able to adaptively manage fire to meet their environmental, cultural and socio-economic objectives.

Non-Indigenous fire managers are sometimes constrained by government requirements to set burn dates based on the Gregorian calendar, coordination of resources and personnel from multiple agencies, and relying on suitable weather conditions on the set date. This means that agreements to burn are often obstructed by one or more of these factors and burning cannot proceed. In contrast, the Yugul Mangi rangers have more adaptability to match their burning practices to cultural Law (through existing kinship and governance systems), the environmental conditions at the time, and can change their plans according to conditions. The Gregorian calendar, which is based on the Northern Hemisphere seasons of summer, autumn, spring and winter, translates poorly to Australia's seasonal conditions. In contrast, Indigenous peoples' seasonal knowledge is intimately related to their country and suits the diversity of environments found in Australia.

For example, the *Yugul Mangi Faiya En Sisen Kelenda* (Figure 3) presented three seasons, given various names in traditional Indigenous languages, aligned to the annual cycle of fire management (Table 2). During the Wet Season, when it rains most days, it is too wet to burn, and an important time to collect bush-tucker fruits. The Dry Season is the right time to burn, when the weather is cool and there are light, south-easterly winds. Biocultural indicators are used to indicate when it is time to start burning, including the appearance of large numbers of dragonflies and the dominant savanna grass species dropping their seeds. This is an important time of transition, when the shrubs that fruited during the Wet Season finish fruiting, and burning should be carefully controlled around these important bush tucker resources. The Build-Up Season is a time of dry, hot weather, when fires have the potential to be larger, more intense and severe. This is the "nugudwan faiya" (wildfire) time, when wildfires threaten to damage important ecological and cultural values. The biocultural indicators for

the transition into the Build-Up Season are the arrival of hot winds, high temperatures and whirlwinds. This is an important time for harvesting magpie geese, freshwater shark and stingray. The rangers stop lighting fires during the Build-Up. The transition from the Build-Up to the Wet Season is marked by biocultural indicators including the appearance of Leichhardt's grasshopper with the first storms, the flowering of the whitewood (shark tree *Atalaya hemiglauca*), wattle (*Acacia* spp.) and paperbark trees, and crocodiles and turtles laying their eggs.

Using cross-cultural knowledge, such as that encapsulated in the *Yugul Mangi Faiya En Sisen Kelenda*, has enabled rangers to improve their fire management. Ansell and Evans [24] found that in comparison to the baseline data (2000–14), during the registered years of the SEALFA2 project (2015–2018), fire management had improved in the SEAL IPA by reducing the total area that was burnt each year, shifting the seasonality of burning from late dry season to early dry season, increasing the patchiness of fires, increasing the area that was considered long unburnt, and reducing the GHG released from burning. This cross-cultural knowledge embraces the complexity of fire management, and allows rangers to make informed decisions for their burning practices. For example, as explained by a senior Yugul Mangi ranger, sometimes the helicopter is booked to burn a certain area of the SEAL IPA within the savanna burning timeframe, but if the grass has not yet turned from green to brown and dropped its seeds, it is not ready to burn, and the helicopter must be rescheduled. The information in the calendar can be also used to train new rangers through the transfer of knowledge contributed by senior rangers and Elders and to explain to external parties why savanna burning operations occur at given times throughout the year.

However, our study showed that there was still a range of views and tensions between Elders and rangers on "right way" application of fire practices and a need for ongoing cultural education and fire awareness in the community. Fires were still occasionally lit late in the dry season which, along with wildfires, can damage ecological and cultural values. For example, participants described incidences where culturally important areas and bush tucker were burnt. Other concerns related to savanna burning were that SEALFA2 had not delivered a profit every year, and the focus on carbon and the use of western tools challenged the balance between western and traditional knowledge and practice. Similar concerns about the carbon economy have been raised by others in nearby Cape York [72]. This further highlights the need to strengthen and grow traditional cultural knowledge, practice, values and intergenerational exchange when adopting western economies, tools and techniques.

In our study, the relationship between fire management and abundance of bush-tucker resources was highlighted as important, with declines in bush-tucker production an ongoing concern for the community. While fire was considered to be an important agent in the maintenance of bush-tucker resources, other impacts such as feral animals (buffalo [*Bubalus bubalis*], pig [*Sus scrofa*], horse [*Equus caballus*], wild cattle [*Bos Taurus* and *B. indicus*]), climatic variation (rainfall), weeds and non-customary harvest were also identified. This correlates with findings by Ens et al. [73] who investigated multiple causes of the decline of the bush-tucker fruit, Djutpi (Jupi, *Antidesma ghaesembilla*), in the SEAL IPA. Ongoing vigilance in protecting important bush-tucker resources from various threats and learning to adapt to new conditions and knowledge will be needed in order to maintain these culturally important resources.

The importance of improving Indigenous burning education and awareness for the community, particularly the younger generation, was consistently emphasized by participants. Participants were concerned that community members were not interested in learning about burning, did not have access to country, or did not have mentors to learn the correct way to burn. We found that community awareness of the rangers' fire management, particularly savanna burning for GHG abatement, was low. The SEAL IPA Plan of Management [57] included the following intended actions: "develop and run culture camps out on country for children and young people" and "run Right-way Fire Workshops in early dry season for students and unemployed youth." In 2019, 63% of SEALFA2 profits were planned to be used for ranger operations and 37% for community development, including culture camps [74], training and facilitating Traditional Owner access to country. As suggested by participants of this

study, the fire and seasons calendar could be used as a teaching resource at these events, in schools and to improve community awareness of fire management.

## 5. Conclusions

Indigenous seasonal calendars from around the world have illustrated how Indigenous knowledge and connection to country provide a deep and intimate understanding of the landscape [11,39–55]. The *Yugul Mangi Faiya En Sisen Kelenda* adds to this growing collection, and aims to improve the evolving fire management practices of Indigenous land managers, while also providing an effective communication tool to increase awareness of Indigenous savanna burning. While Prober et al. [41] lamented that seasonal knowledge is not yet well embedded in natural resource management and recommended investigation of how its applications might be improved, we have demonstrated a way in which a seasonal calendar can be used for multiple purposes.

Savanna burning has provided Indigenous peoples with the economic support to care for country through the fundamental practice of fire management. Globally, recognition of Indigenous knowledge and management of fire is growing, but real action on policy and on ground works is limited [75]. The Yugul Mangi rangers are an example of an Indigenous group who have transcended the barriers to Indigenous leadership in fire management. Documenting their local and Traditional Ecological Knowledge through the *Yugul Mangi Faiya En Sisen Kelenda* has allowed them to share their knowledge and explain their world-view related to fire management. Many Indigenous groups around the world aspire to increase their participation in fire management and have their knowledge recognized and appreciated [76]. We recommend the development of fire and seasons calendars as a practical and educational activity that could be relevant to many of these communities and help avoid potential misunderstandings between Indigenous and non-Indigenous fire managers. This could drive an increase in the uptake of adaptive and locally attuned Indigenous fire management to alleviate poverty, cherish Indigenous knowledge, fight climate change and restore ecosystems in many locations across the globe.

**Author Contributions:** All authors have read and agree to the published version of the manuscript. Conceptualization, E.E, O.C. and Y.M.R.; methodology, E.E., M.M.; validation, E.E. and N.R.; investigation, Y.M.R M.M. and E.E.; resources, M.M.; data curation, M.M.; writing—original draft preparation, M.M.; writing—review and editing, E.E., N.R., Y.M.R O.C. and M.M.; visualization, E.E. and N.R.; supervision, N.R. and E.E.; project administration, M.M. and Y.M.R.; funding acquisition, O.C., M.M. All authors have read and agreed to the published version of the manuscript.

**Funding:** This research was funded by University of New England, Firesticks Project & Rural Fire Service NSW.

**Acknowledgments:** The authors would like to acknowledge the Traditional Owners of South East Arnhem Land and all Indigenous people, past, present and future, who have cared for and shared their knowledge of country and culture. The authors would like to acknowledge Catherine MacGregor for assistance with map preparation, Ngukurr Language Centre for assistance with Indigenous languages and Kerry Hardy for graphic design of the *Yugul Mangi Faiya En Sisen Kelenda*.

**Conflicts of Interest:** The authors declare no conflict of interest. The funders had no role in the design of the study; in the collection, analyses, or interpretation of data; in the writing of the manuscript, or in the decision to publish the results.

## Appendix A

*Semi-structured interview questions*
*Field work June 2016:*

- What are the seasons here?
- When is the hot/cold time?
- What is the weather like?
- What bush tucker do you eat?
- When do you eat it?
- Do any plants or animals tell you when the seasons are changing or that animals are ready to eat?

- When do you burn?
- When should you not burn?
- How do you know if it is the right time to burn?
- Who does the burning?
- How is the burning going?
- Do you go out burning?
- How did you learn how to burn?
- How did the old people burn?
- Why did the old people burn?
- Why do you burn today?
- How do you burn now?
- How is burning different today to in the past?
- When you go out burning how does it make you feel?
- Do kids learn about burning?
- Can you tell me about carbon farming/savanna burning?

*Follow up questions for field work June 2019:*

- Is the information in the Fire & Seasons calendar correct?
- Are you happy to have your quote/photo/information used in the calendar, paper and thesis publications?
- How is burning in SEAL IPA going?
- How is SEALFA progressing?
- What have you learned/experienced in the first three years of SEALFA?
- What benefits have you seen from SEALFA?
- How does SEALFA affect:
- Resources for rangers and the community?
- The relationship between rangers and Traditional Owners?
- What are the issues that are causing tension?
- Are you able to undertake traditional fire management as part of SEALFA?
- Were you able to undertake traditional fire management before SEALFA?
- Do Traditional Owners go out burning with rangers?
- Do you think this calendar will help? How?
- How will you use this calendar?
- How can we improve the draft calendar?
- What things need to be done to improve fire management overall?

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
