# Peer review of "Indigenous Knowledge and Seasonal Calendar Inform Adaptive Savanna Burning in Northern Australia"

_sustainability, doi:10.3390/su12030995_

Round 1
Reviewer 1 Report
Indigenous Knowledge and seasonal calendar inform adaptive savanna burning in northern Australia
General comments
I have read the manuscript entitled “Indigenous Knowledge and seasonal calendar inform adaptive savanna burning in northern Australia”. The article deals with the importance of indigenous knowledge, in this specific case seasonal calendars, as management tool to help in the use of prescribe burns. The article is interesting as there are not many examples of indigenous knowledge applied in forest fires prevention plans. However, the article needs some improvement specially in the introduction and results presentation.
Introduction starts with a vague and general statement about indigenous role in managing natural hazards. This statement is only partially true, there are indigenous groups that use fire to clear land for agriculture, this is especially true for central America and Asia. Another classic example is eastern island were people almost starve to death because of overexploiting natural resources. I suggest that you focus and state clearly that the you are referring to activities that are beneficial and that these activities are heavily influenced by the cosmovision of each indigenous group. In addition, there is barely any information about why these prescribe burns are necessary in the Australian bush context “Less fuel, lower the probability of fire”. Also, there no explanation of what a seasonal calendar is and their link with prescription fires. For a person reading this without any knowledge about the Australian bush fires will not be able to connect with the article. I think is important to state that Europeans did not see the value in doing prescribe burns. Prescribe burns were seeing as a curiosity and there were not valued as valid management strategy until recently. Then you could start building your case around the importance of prescribe burns and all the initiatives that are in place in Australia. Please add benefits of burning in ecological and economical terms. Avoid bullet points in the introduction try to add all those points using a paragraph structure. You can add this information in a paragraph defining what a prescribe burn is and the pros and cons of these activities. The objectives are not clearly stated in the introduction, please rewrite the last section of the introduction.
About the methods the small sample sizes and underrepresentation of some gender and roles can lead to generalizations when there are not enough interviews to do so. In this sense I suggest adding a paragraph in the discussion section stating that results must interpret with caution avoiding generalizations due to the small sample size. In addition, it is not clear if the opinion of the different community members were weighted according to their sample size or if the gender/role groups were analyzed separately.
In the results sections some of statements presented are out of place and sometimes are difficult to follow. For example, lines 173-174, the article is mainly focus in prescribed burns, but you mention “Fire was used as an important tool to signal between groups” which is not connected with the previous statements. Please avoid adding information that is not relevant to the objectives of the article. Please when you are citing a person statement start with little context avoiding just copying the person statement directly.
The discussion section is disjointed, disorganized and is missing crucial information. For example, it is not clear the link between indigenous knowledge and seasonal calendar with the burn practices. In addition, the identification of important practices that can be replicated elsewhere are missing. I believe that the discussion should be focus on the link between indigenous knowledge and burn practices, how and which of these practices can help in preventing forest fires.
I believe that this article has potential but there are some major changes that are needed to make this article more robust. In consequence, I recommend that the article should be considered after major changes are made.
Author Response
Reviewer 1: General comments
I have read the manuscript entitled “Indigenous Knowledge and seasonal calendar inform adaptive savanna burning in northern Australia”. The article deals with the importance of indigenous knowledge, in this specific case seasonal calendars, as management tool to help in the use of prescribe burns. The article is interesting as there are not many examples of indigenous knowledge applied in forest fires prevention plans. However, the article needs some improvement specially in the introduction and results presentation.
Introduction starts with a vague and general statement about indigenous role in managing natural hazards. This statement is only partially true, there are indigenous groups that use fire to clear land for agriculture, this is especially true for central America and Asia. Another classic example is eastern island were people almost starve to death because of overexploiting natural resources. I suggest that you focus and state clearly that the you are referring to activities that are beneficial and that these activities are heavily influenced by the cosmovision of each indigenous group. In addition, there is barely any information about why these prescribe burns are necessary in the Australian bush context “Less fuel, lower the probability of fire”. Also, there no explanation of what a seasonal calendar is and their link with prescription fires. For a person reading this without any knowledge about the Australian bush fires will not be able to connect with the article. I think is important to state that Europeans did not see the value in doing prescribe burns. Prescribe burns were seeing as a curiosity and there were not valued as valid management strategy until recently. Then you could start building your case around the importance of prescribe burns and all the initiatives that are in place in Australia. Please add benefits of burning in ecological and economical terms. Avoid bullet points in the introduction try to add all those points using a paragraph structure. You can add this information in a paragraph defining what a prescribe burn is and the pros and cons of these activities. The objectives are not clearly stated in the introduction, please rewrite the last section of the introduction.
Response 1: We have added two paragraphs at the beginning of the Introduction that provide further explanation on Indigenous fire management history, practices and how it relates to their cosmovision. We have added international references that provide background on wildfires as a global issue (including references related to Europe and their burning methods). The ecological and economic benefits are described in the 8th paragraph of the Introduction, lines 89 – 104. Bullet points have been removed. The last section of the Introduction has been rewritten, with clear definition of the knowledge gap and objectives of the study.
About the methods the small sample sizes and underrepresentation of some gender and roles can lead to generalizations when there are not enough interviews to do so. In this sense I suggest adding a paragraph in the discussion section stating that results must interpret with caution avoiding generalizations due to the small sample size. In addition, it is not clear if the opinion of the different community members were weighted according to their sample size or if the gender/role groups were analyzed separately.
Response 2: as suggested by another reviewer, we have simplified and reduced the Methods and Results sections. The quantitative analysis of data has been removed. We have explained in the Methods why there was a small sample size. The representation of the gender and roles is a reflection of demographics of the population.
In the results sections some of statements presented are out of place and sometimes are difficult to follow. For example, lines 173-174, the article is mainly focus in prescribed burns, but you mention “Fire was used as an important tool to signal between groups” which is not connected with the previous statements. Please avoid adding information that is not relevant to the objectives of the article. Please when you are citing a person statement start with little context avoiding just copying the person statement directly.
Response 3: We have rewritten most of the results section and made it simpler, shorter and easier to understand. We have made sure that any quotes are placed in context and explained in the text. We have deleted the statement “Fire was used as an important tool to signal between groups”.
The discussion section is disjointed, disorganized and is missing crucial information. For example, it is not clear the link between indigenous knowledge and seasonal calendar with the burn practices. In addition, the identification of important practices that can be replicated elsewhere are missing. I believe that the discussion should be focus on the link between indigenous knowledge and burn practices, how and which of these practices can help in preventing forest fires.
Response 4: we have rewritten the Discussion to focus more on the link between indigenous knowledge and the seasonal calendar with the burn practices (see paragraphs 1 – 4 of the Discussion). We have identified important practices that can be replicated elsewhere and tried to give the paper a more international focus.
I believe that this article has potential but there are some major changes that are needed to make this article more robust. In consequence, I recommend that the article should be considered after major changes are made.
Response 5: Thank you very much Reviewer 1 for your thoughtful responses to our paper. We have made major changes to the paper to try to customise it to an international audience and to explain the research findings that could have applications elsewhere. We have provided more background information and tried to focus on the key issues of why do prescription burns in Australia, what is a seasonal calendar, what is the link between Indigenous Knowledge, seasonal calendar and burn practices and how are fire activities influenced by an Indigenous group's cosmovision. We hope that you find our changes satisfactory and look forward to sharing the findings of our research.
Please also see attached word doc for all responses to reviewers.
Reviewer 2 Report
I felt that this work would be useful as a remedy especially in the context of large wildfires across Australia and the United States.

Author Response
Reviewer 2:
Basic Reporting- The knowledge gap was not explained well.
- The objective was also not emphasized well.
- Really long sentences, and long paragraph. I would suggest author to shorten the sentences and the paragraphs and make a connection from one paragraph to another paragraph so that the reader like me do not get lost.
- Overall the paper emphasized on the seasonal calendar emphasized by the indigenous people which for me as person working in wildfire management looks promising and have a loads of good information that can be used in the scientific community.
Response 1: The last section of the Introduction has been rewritten, with clear definitions of the knowledge gap and objectives of the study. Much of the paper has been rewritten and customised to an international audience. The sentences and paragraphs have been shortened and their key messages made clearer. Thank you for your encouragement regarding the promising nature of our paper, we hope through our improvements and major revisions it can be shared widely with the scientific community.
Experimental DesignThe research article aligns with the scope of the journal. As a reader, I felt that the research question was not defined well although the gap is mentioned but not explained well. The methodology along with the result sections need to be revised thoroughly. A lot of detail were presented in the both of the sections which were not needed. So, I would urge the author to trim these section.
Response 2: We have added clear definitions of the knowledge gap and objectives of the study. We have simplified and reduced the Methods and Results sections. The quantitative analysis of data has been removed. We have revised the Methods and Results sections and removed a lot of unnecessary detail and trimmed these sections.
Validity of findingThe way the result is presented, the author had worked in detailed to provide information on the fire management done by the Indigenous people. I would recommend to make a list of the important take home messages from the Result as well as Discussion section and then described that. Because the sections are too long and I am lost as I read through paragraph.
Response 3: acknowledging that these sections were too long, we have shortened and simplified them. We have focused on the key take home messages of the research as suggested by you and the other reviewers. We have removed extraneous information so that we can focus on the core messages.
Please revised the section as the edits provided in the attachment.
General comment on manuscript:This paper needs to highlight the why Indigenous Knowledge and seasonal calendar inform adaptive savanna burning in northern Australia. Every section is lengthy so need to trim it.
The author should pick up the main point and explained on it.
Author needs to present a more comprehensive view of the literature surrounding the use of the knowledge and the seasonal calendar on adaptive savanna burning. I recommend that the author works on each feedback the reviewer provided to make this paper better. Some Major revisions are necessary before publication.
Response 4: we have rewritten most of the paper and made major changes in order to accept the suggestions made by reviewers. We have added several paragraphs to the Introduction to provide more background and a comprehensive literature review on key issues such as Indigenous knowledge, fire management and seasonal calendars. We have removed a lot of the extraneous information (e.g. about the detailed practices of savanna burning) from the Introduction, Results and Discussion. We have tried to simply the paper so that are core messages are clear and the paper is not too long.
Major comments:
Each section is long. The author make sure to work concising the paper. It is very long. Some of the section from the Introduction section can be integrated into material and methodology section which is addressed in the attached file. Even for the result section, some of the paragraph can be combined together. The result section is too long and I would suggest the author to get the main take home message in the section. The author should pick up the important figures and tables in the manuscript. And for figure 3, the author can change the figure as shown below (FE, ME and other categories can be done as red, tan and lime) so that it is easy to visualize.
Response 5: We have moved the information on the study area from the Introduction to the Methods. We have cut 2 sub-sections out of the Results, as well as Table 2 and Fig 3, making this Results section much shorter. We have changed the text to make it easier to understand. Interpretation of the Figures and Tables are discussed in the Discussion.
Please look on the edits on the attachment.
The attachment contains specific comments and edits.
Line 24: Could not get the research gap for this study?
Response 6: Research gap has been identified in the last paragraph of the Introduction
Line 38-40: Please work on these sentence.
Response 7: This sentence have been changed as part of the rewrite
Line 49-52: It is very long and hard to understand.
Response 8: This sentence has been broken into two and made clearer.
Line 76-79: These two sentence can be made one.
Response 9: These sentences have been changed as part of the rewrite
Line 83-85: Can go to the Methodology section.
Response 10: Has been moved to the Methodology section
Line 223: Cite the paper as it provides information on how canopy height, NDVI and the day of year were used to estimate fuel moisture content which ultimately helped in the fire management aspects (from remote sensing aspects) (https://www.researchgate.net/profile/Sonisa_Sharma/publication/323506897_Nondestructive_Estimation_of_Standing_Crop_and_Fuel_Moisture_Content_in_Tallgrass_Prairie/links/5ae8d28fa6fdcc03cd8f80ef/Nondestructive-Estimation-of-Standing-Crop-and-Fuel-Moisture-Content-in-Tallgrass-Prairie.pdf)
Response 11: Thank you for the suggestion, that looks like a really interesting paper and we would like to learn more. Unfortunately as part of the process to remove some of the technical details from the paper to make it shorter and clearer, we have removed similar references. We would be interested to know more and to share this research in future papers.
Table 3 should be short and understandable.
Response 12: we have made Table 3 as concise as we possibly can. It is difficult to reduce the information any more due to its complexity. Other published scientific papers presenting seasonal knowledge have used similar tables, for example, see Table 3 in: Balehegn, M., Balehey, S., Fu, C. & Liang, W. Indigenous Weather and Climate Forecasting Knowledge among Afar Pastoralists of North Eastern Ethiopia: Role in Adaptation to Weather and Climate Variability. Pastoralism 2019, 9.
General comments: I felt that this work would be useful as a remedy especially in the context of large wildfires across Australia and the United States.
Response 13: Thank you! We appreciate you taking the time to help us to improve our paper and for recognising the potential uses of our research.
We have also attached a word document showing all responses to the 4 reviewers of our paper.
Reviewer 3 Report
The manuscript titled "Indigenous knowledge and seasonal calendar inform adaptative savanna burning in northern Australia" wants to ellaborate some guidelines for improving the indigenous ranger's fire management planning. However, the manuscript does not correctly descrive an adequate calendar season.
The goal of the manuscript is not clear in the introduction. In the present form, the study seems a description and justification of the SEALFA2 project.
In my opinion, the discussion section mainly descrives the results of the second part of the query form of 2019 (for example, only 4 citations have been included in this section). In my opinion, the introduction and the discussion section should also include a comparison between indigenous and non-indigenous fire management due to fire management changed from 1860 with the colonization by Europeans (Lines 47-49).
Queries have been done to 21 persons in the Ngukurr town. How many people live in the area? Do the opinion of the 10 interviewees which did the query in both years changed from the first to the second interviews? Authors mention in the manuscript that rangers and elder people have been interviewed and authors separe their query answers in the results section. However, I am curious to know more about why most of the elders comments are negative while rangers expressed more positive comments or, in other words, which fire management/calendar is better. Why answers changed from 2016 to 2019? Any answer (or group of answers) is repeated between years (which difficults their comparison).
Please, descrive the ecological calendar and compare it to the fire management calendar of the country (for example) in order to compare which calendar is the most appropiate. It is really interesting to see that the traditional fire management period differs to the rangers' ones.
I also suggest to move the description of the study area to the Materials and Methods section and to include some other interesting information such as the distribution of population, traditional land uses, etc, which can help in understanding better the relation of the population with the use of fire (as descrived in: https://www.nlc.org.au/uploads/pdfs/SEAL-IPA-PoM-V3.4.pdf).
Author Response
Reviewer 3:
The manuscript titled "Indigenous knowledge and seasonal calendar inform adaptative savanna burning in northern Australia" wants to ellaborate some guidelines for improving the indigenous ranger's fire management planning. However, the manuscript does not correctly descrive an adequate calendar season.
Response 1: We have changed the paper to include a full description of the annual cycle and seasons in the Discussion (3rd paragraph of the Discussion). These calendar seasons are also represented in Fig 3 and Table 2.
The goal of the manuscript is not clear in the introduction. In the present form, the study seems a description and justification of the SEALFA2 project.
Response 2: Much of the Introduction has been rewritten to provide more background and a comprehensive literature review on Indigenous knowledge, fire management and seasonal calendars. The last section of the Introduction has been rewritten, with clear definition of the knowledge gap and objectives of the study. The paper has been changed to focus on the key issues of Indigenous knowledge, fire management and seasonal calendars and reduced the amount of text regarding a description and justification of the SEALFA project. We have presented both positive and negative opinions related to the SEALFA project.
In my opinion, the discussion section mainly descrives the results of the second part of the query form of 2019 (for example, only 4 citations have been included in this section). In my opinion, the introduction and the discussion section should also include a comparison between indigenous and non-indigenous fire management due to fire management changed from 1860 with the colonization by Europeans (Lines 47-49).
Response 3: We have rewritten most of the Discussion. We have shortened the paper and removed the comparison section related to sentiment in 2016 and 2019. We have added comparisons of Indigenous and non-Indigenous fire management in the Introduction (4th paragraph) and Discussion (2nd paragraph of Discussion). We have included information about how fire management changed prior to, and following, colonisation by Europeans in the Introduction (4th paragraph of Introduction).
Queries have been done to 21 persons in the Ngukurr town. How many people live in the area? Do the opinion of the 10 interviewees which did the query in both years changed from the first to the second interviews? Authors mention in the manuscript that rangers and elder people have been interviewed and authors separe their query answers in the results section. However, I am curious to know more about why most of the elders comments are negative while rangers expressed more positive comments or, in other words, which fire management/calendar is better. Why answers changed from 2016 to 2019? Any answer (or group of answers) is repeated between years (which difficults their comparison).
Response 4: We have added information in the Methods to provide more background on the demographics of the study area. In order to simply the paper and improve clarity (as recommended by other reviewers), we have removed the section comparing sentiment from 2016 – 2019. This has been deleted from the Methods, Results and Discussion section.
Please, descrive the ecological calendar and compare it to the fire management calendar of the country (for example) in order to compare which calendar is the most appropiate. It is really interesting to see that the traditional fire management period differs to the rangers' ones.
Response 5: We have changed the paper to include a full description of the annual cycle and seasons in the Discussion (3rd paragraph of the Discussion). These calendar seasons are also represented in Fig 3 and Table 2. The rangers interviewed were Indigenous, as were the Elders. We have compared the Indigenous calendar with the approach of non-Indigenous fire managers who generally use the Gregorian calendar for their planning (Discussion paragraph 2).
I also suggest to move the description of the study area to the Materials and Methods section and to include some other interesting information such as the distribution of population, traditional land uses, etc, which can help in understanding better the relation of the population with the use of fire (as descrived in: https://www.nlc.org.au/uploads/pdfs/SEAL-IPA-PoM-V3.4.pdf).
Response 6: We have moved the description of the study area to the Materials and Methods sections and added more description of the demographics of the region.
Thank you Reviewer 3 for taking the time to read and improve our manuscript. We hope that you find our major changes to paper to be adequate, and we look forward to sharing our research findings with the international research community.
We have also attached a document showing our responses to all 4 reviewers.
Reviewer 4 Report
This manuscript presented the usefulness and importunateness of indigenous knowledge in savanna burning monitoring and management. The authors took South East Arnhem Land Indigenous Protected Area (SEAL IPA) as a case study, with actions of interviews with Elders and rangers during 2016 and 2019 to investigate the indigenous knowledge and seasonal calendar for best practice, adaptive management of fire. This manuscript is based on the humanities and social sciences, focusing on the use of interviews and surveys to demonstrate and summarize the consequence of the scheme. The ultimate goal is to provide reasonable guidance for fire management at regional or global scale. The structure of the manuscript is simple and straightforward for readers to understand. Given the important role of savanna fire information in conducting fire management activities, I think this study is interesting.
Some general comments:
- A total sample of twenty-one participants is quite small, do you think it is sufficient to conduct your research and make the conclusion? Particularly, only 10 of 21 were interviewed in both 2016 and 2019, which leads to statistically inconsistency.
- From the manuscript, indigenous fire burning is an effective way for the fire management. Just curious, how do the rangers control the size, intensity, and location of the fire? If the intensity is out of control, such as, not only burn the grass-land but also burn the tree canopy, even the bush trucker, what will they do to reduce losses?
- Indigenous knowledge is critical for fire management. What is the role of modern technologies (e.g., satellite remote sensing and GIS) in Indigenous knowledge-oriented fire management?
- Is this method applicable in other fire-prone areas and countries? For example, the Mediterranean coast, Portugal, Spain, California of the United States, etc.?
Author Response
Reviewer 4:
This manuscript presented the usefulness and importunateness of indigenous knowledge in savanna burning monitoring and management. The authors took South East Arnhem Land Indigenous Protected Area (SEAL IPA) as a case study, with actions of interviews with Elders and rangers during 2016 and 2019 to investigate the indigenous knowledge and seasonal calendar for best practice, adaptive management of fire. This manuscript is based on the humanities and social sciences, focusing on the use of interviews and surveys to demonstrate and summarize the consequence of the scheme. The ultimate goal is to provide reasonable guidance for fire management at regional or global scale. The structure of the manuscript is simple and straightforward for readers to understand. Given the important role of savanna fire information in conducting fire management activities, I think this study is interesting.
Response 1: Thank you for taking the time to read our paper and provide feedback. We are glad that your think our study is interesting.
Some general comments:
- A total sample of twenty-one participants is quite small, do you think it is sufficient to conduct your research and make the conclusion? Particularly, only 10 of 21 were interviewed in both 2016 and 2019, which leads to statistically inconsistency.
Response 2: We have provided an explanation of the demographics of Ngukurr and the study site in the Methods which we hope will justify the sample size of twenty-one participants. In order to shorten the paper and make it clearer, we have removed the comparison and quantitative analysis of participants’ responses from the paper.
- From the manuscript, indigenous fire burning is an effective way for the fire management. Just curious, how do the rangers control the size, intensity, and location of the fire? If the intensity is out of control, such as, not only burn the grass-land but also burn the tree canopy, even the bush trucker, what will they do to reduce losses?
Response 3: the rangers control the fire by undertaking it under suitable conditions during particular times of the year (this is described in the Discussion, paragraph 3). By undertaking the majority of their burning during the early dry season, the fires are lower intensity and less severe. This reduces the fuel load and creates fire breaks which reduces the likelihood of destructive wildfires occurring the late dry season (this is explained in the 3rd paragraph of the Methods and also in the 3rd paragraph of the Discussion)
- Indigenous knowledge is critical for fire management. What is the role of modern technologies (e.g., satellite remote sensing and GIS) in Indigenous knowledge-oriented fire management?
Response 4: this is explained in the Results section 3.2 and also the first paragraph of the Discussion
- Is this method applicable in other fire-prone areas and countries? For example, the Mediterranean coast, Portugal, Spain, California of the United States, etc.?
Response 5: Yes it is! We have discussed the international applications in the Conclusion section.
We have also attached a word showing all the changes that we have made to comment from the 4 reviewers.
Thank you Reviewer 4 for taking the time to read our paper and help us to improve it.
Round 2
Reviewer 2 Report
The paper had improved a lot although some of the feedback that was asked was not implemented. I have attached the pdf file where the feedback has been attached.

Author Response
Thank you for providing further comments on our manuscript. Sorry we overlooked your previous comments, we could not find them. We hope we have addressed all of your concerns in our responses (in red text) below:
Comments from PDF:
The term nation-state is correct in this instance (not country’s states) and so we have kept the word ‘nation-states’ We have deleted ‘however’ and kept ‘while’ Is there any reference for it? what is fire-scar map? That's why I asked you to refer that paper which used canopy height, day of year (seasonal calendar) and remotely sensed data. The author can just say "Remote sensed data have proved an useful tool to improve fire management practices (Sharma et al. 2018)". So delete this sentence and keep this one. This might give more evidence on fire management practices.We have changed the sentence to the following:
Remote-sensed data have proved a useful tool to improve fire management practices [67] and remotely-sensed fire-scar maps are used to plan and review fire management.
Bring a opening sentence and then use bullet point.We have added an opening sentence, then used bullet points:
The Yugul Mangi Rangers formally commenced savanna burning in 2016. During the first 3 years of operation, participants shared that savanna burning has:
provided income for the rangers to buy work-related resources, including gators, trailer, vehicles, fuel, helicopter time, firefighting uniforms, drip torches, matches, Raindance machine and pay for casual employees supported rangers in their fire management by improving engagement with western technology such as GPS, helicopters and satellite mapping whilst still using Indigenous ways of burning reduced wildfires by increasing early dry season burning. Merge two lines- two lines have been merged I still feel this table is way too long. A similar paper on Traditional Ecological Knowledge and seasonal calendar, also published in Sustainability, had a table that was 3 pages long (Table 3), see: Yang, H.; Ranjitkar, S.; Zhai, D.; Zhong, M.; Goldberg, S.D.; Salim, M.A.; Wang, Z.; Jiang, Y.; Xu, J. Role of Traditional Ecological Knowledge and Seasonal Calendars in the Context of Climate Change: A Case Study from China. Sustainability 2019, 11, 3243. We have made our Table 3 as concise as we possibly can. It is difficult to reduce the information any more due to its complexity. This is only a short excerpt (presented as text) of the total information that is presented in Fig 3. The nature of the information means that it is complex and cannot be simplified any further. This is also shown in the other papers we have mentioned here that also have large tables. Other published scientific papers presenting seasonal knowledge have used similar tables, for example, see Table 3 in: Balehegn, M., Balehey, S., Fu, C. & Liang, W. Indigenous Weather and Climate Forecasting Knowledge among Afar Pastoralists of North Eastern Ethiopia: Role in Adaptation to Weather and Climate Variability. Pastoralism 2019, 9.Reviewer 3 Report
The manuscript has been improved and currently includes most of reviewers suggestions. The only comment I have is related to the advantages (or improvements) of the new seasonal calendar compared to the traditional ones. Can authors list some of them?
Minor comments:
- Lines 287-288: Please, remove the line break.
Author Response
Thank you for providing further comments on our manuscript. We have attempted to address your concerns, please see our responses below in red text:
The manuscript has been improved and currently includes most of reviewers suggestions. The only comment I have is related to the advantages (or improvements) of the new seasonal calendar compared to the traditional ones. Can authors list some of them?
We have added additional information to compare the Indigenous seasonal calendar and Gregorian calendar (see below). The benefits of the Indigenous calendar are also explained throughout the Discussion:
Non-Indigenous fire managers are sometimes constrained by government requirements to set burn dates based on the Gregorian calendar, coordination of resources and personnel from multiple agencies, and relying on suitable weather conditions on the set date. This means that agreements to burn are often obstructed by one or more of these factors and burning cannot proceed. In contrast, the Yugul Mangi Rangers have more adaptability to match their burning practices to cultural Law (through existing kinship and governance systems), the environmental conditions at the time, and can change their plans according to the conditions. The Gregorian calendar, which is based on the Northern Hemisphere seasons of summer, autumn, spring and winter, translates poorly to Australia’s seasonal conditions. In contrast, Indigenous peoples’ seasonal knowledge is intimately related to their Country and suits the diversity of environments found in Australia.
For example, the Yugul Mangi Faiya En Sisen Kelenda (Figure 3) presented three seasons, given various names in traditional Indigenous languages, aligned to the annual cycle of fire management (Table 2)...
Minor comments:
- Lines 287-288: Please, remove the line break. DONE
Reviewer 4 Report
The authors have appropriately replied to my comments and questions.
Author Response
Thank you for your comments and suggestions, we are grateful for your help in improving our manuscript.